# Restrictive Cardiomyopathy is Caused by a Novel Homozygous Desmin (*DES*) Mutation p.Y122H Leading to a Severe Filament Assembly Defect

**DOI:** 10.3390/genes10110918

**Published:** 2019-11-11

**Authors:** Andreas Brodehl, Seyed Ahmad Pour Hakimi, Caroline Stanasiuk, Sandra Ratnavadivel, Doris Hendig, Anna Gaertner, Brenda Gerull, Jan Gummert, Lech Paluszkiewicz, Hendrik Milting

**Affiliations:** 1Erich and Hanna Klessmann Institute for Cardiovascular Research & Development (EHKI), Heart and Diabetes Center NRW, University Hospital of the Ruhr-University Bochum. Georgstrasse 11, D-32545 Bad Oeynhausen, Germany; cstanasiuk@hdz-nrw.de (C.S.); sratnavadivel@hdz-nrw.de (S.R.); agaertner@hdz-nrw.de (A.G.); jgummert@hdz-nrw.de (J.G.); 2Otto-von-Guericke University, Magdeburg, Universitätsplatz 2, D-39106 Magdeburg, Germany; s.pourhakimi@gmail.com; 3Institute for Laboratory and Transfusion Medicine, Heart and Diabetes Center NRW, University Hospital of the Ruhr-University Bochum, Georgstrasse 11, D-32545 Bad Oeynhausen, Germany; dhendig@hdz-nrw.de; 4Department of Cardiac Sciences, Libin Cardiovascular Institute of Alberta, University of Calgary, Calgary, AB T2N4Z6, Canada; gerull_b@ukw.de; 5Comprehensive Heart Failure Center and Department of Internal Medicine I, University Hospital Würzburg, D-97080 Würzburg, Germany; 6Department of Cardio-Thoracic Surgery, Heart and Diabetes Center NRW, University Hospital of the Ruhr-University Bochum, Georgstrasse 11, 32545 Bad Oeynhausen, Germany; lpaluszkiewicz@hdz-nrw.de

**Keywords:** cardiovascular genetics, restrictive cardiomyopathy, desmin, intermediate filaments, desmin-related myopathy, cardiomyopathy, desminopathy

## Abstract

Here, we present a small Iranian family, where the index patient received a diagnosis of restrictive cardiomyopathy (RCM) in combination with atrioventricular (AV) block. Genetic analysis revealed a novel homozygous missense mutation in the *DES* gene (c.364T > C; p.Y122H), which is absent in human population databases. The mutation is localized in the highly conserved coil-1 desmin subdomain. In silico, prediction tools indicate a deleterious effect of the desmin (*DES*) mutation p.Y122H. Consequently, we generated an expression plasmid encoding the mutant and wildtype desmin formed, and analyzed the filament formation in vitro in cardiomyocytes derived from induced pluripotent stem cells and HT-1080 cells. Confocal microscopy revealed a severe filament assembly defect of mutant desmin supporting the pathogenicity of the *DES* mutation, p.Y122H, whereas the wildtype desmin formed regular intermediate filaments. According to the guidelines of the American College of Medical Genetics and Genomics, we classified this mutation, therefore, as a novel pathogenic mutation. Our report could point to a recessive inheritance of the *DES* mutation, p.Y122H, which is important for the genetic counseling of similar families with restrictive cardiomyopathy caused by *DES* mutations.

## 1. Introduction

Mutations in *DES*, encoding the muscle specific intermediate filament protein desmin, can cause myopathies, as well as different cardiomyopathies [1]. Of note, the spectrum of cardiac phenotypes associated with *DES* mutations ranges from dilated (DCM, MIM #604765) [2,3], arrhythmogenic (ACM, MIM #107970) [4,5], noncompaction (NCCM, MIM #604169) [6], hypertrophic (HCM, MIM #115197) [7] and, in rare cases, also restrictive cardiomyopathies (RCM, MIM #115210) [8]. However, it is currently unknown, why phenotypes caused by *DES* mutations are diverse and include skeletal and cardiac myopathies. Different cardiomyopathies have been observed even within the same family [9]. Presumably, different phenotypes can change over time and cardiac or skeletal muscle involvement might begin at different time points during disease progression. In addition to mutations in *DES*, RCM-associated mutations have been currently described in more than 15 further genes such as *ACTC1* (cardiac actin) [10], *ACTN2* (cardiac actinin-2) [11], *TTN* (titin) [12], *FLNC* (filamin-C) [13], and *CRYAB* (αB-crystallin) [14]. 

Most of the described pathogenic *DES* mutations are heterozygous missense or small in-frame deletion mutations. Cell transfection experiments, as well as analysis of recombinant mutant desmin, have revealed an abnormal aggregation which indicates that the formation of a “poison protein” might be involved in *DES*-associated cardiomyopathies [15,16]. In consequence, additional desmin-associated proteins are involved in the aggregate formation affecting the cardiomyocyte structure [17], however, additional pathogenic *DES* mutations are known which do not induce aggregate formation but change, for example, the nanomechanical properties of the intermediate filaments [18]. The desmin molecules form coiled-coil dimers, which assemble into antiparallel tetramers and unit length filaments (ULFs) [19,20]. The ULFs anneal longitudinally into regular intermediate filaments, which connect different cell organelles such as desmosomes, costameres, Z-bands, mitochondria, and the nuclei [21].

In this study, we identified a novel homozygous missense *DES* mutation p.Y122H (c.364T > C) by next-generation sequencing (NGS) in a patient with severe RCM. Interestingly, a different mutation at the same amino acid position (p.Y122C) has been recently described without further functional analysis in a patient with ACM [22]. Functional analysis of both mutations (p.Y122H and p.Y122C) using HT-1080, SW13 cells, and human induced pluripotent stem cell (iPSC) derived cardiomyocytes in combination with confocal microscopy revealed an abnormal desmin aggregation supporting their pathogenicity.

## 2. Materials and Methods 

### 2.1. Ethitical Approval

This study follows the principles of the Declaration of Helsinki [23] and was approved by the local ethics committee (Ruhr University Bochum, Bad Oeynhausen, Germany, Reg. No. 27.1/2011). Participating patients gave their written consent to the study and agreed to publish the results in anonymous form.

### 2.2. Genetic Analysis

Genomic DNA was isolated from blood by the High Pure PCR Template Preparation Kit (Roche Life Sciences, Prenzberg, Germany) according to the manufacturer’s instructions. The TruSight Cardio gene panel, covering 174 cardiomyopathy-associated genes, was used in combination with the MiSeq system (Illumina, San Diego, CA, USA) for NGS according to the manufacturer’s instructions. Sanger sequencing (Macrogen, Amsterdam, The Netherlands) was used for verification of the *DES* mutation p.Y122H. 

### 2.3. Molecular Modeling

The molecular modeling of the desmin fragment was performed using a template structure of the homologous protein vimentin (3S4R, Protein Structure Databank, [24]) using the Swiss-Model Server (https://swissmodel.expasy.org). The PyMOL Molecular Gaphics System (Schrödinger LCC, New York, NY, USA) was used for visualization. 

### 2.4. Plasmid Generation 

The plasmid pmRuby-N1-*DES* has been previously described by [2]. The QuikChange Lightning Kit (Agilent Technologies, Santa Clara, CA, USA) was used to insert the missense *DES* mutations p.Y122H and p.Y122C using appropriate oligonucleotides. Protein coding sequences of all generated plasmids were verified by Sanger sequencing (Macrogen, Amsterdam, The Netherlands). 

### 2.5. Cell Culture 

The HT-1080 cells were received from the Deutsche Sammlung von Mikroorganismen und Zellkulturen (DSMZ, #ACC315, Braunschweig, Germany) and SW13 cells were received from ATCC (Manassas, USA). The HT-1080 and SW13 cells were cultured in Dulbecco’s Modified Eagle Medium (DMEM, Thermo Fisher Scientific, Waltham, MA, USA) supplemented with 10% fetal calf serum and penicillin/streptomycin. Both cell lines did not express endogenous desmin [25]. Cells were grown in µSlide chambers (Ibidi, Martinsried, Germany). Lipofectamine 3000 (Thermo Fisher Scientific) was used according to the manufacturer’s instruction for cell transfection. These transfection experiments were performed in triplicate or more. The human iPSCs from a healthy donor (NP00040-8, UKKi011-A, European Bank for induced pluripotent Stem Cells, https://ebisc.org/) were kindly provided by Dr. Tomo Saric (University of Cologne) and were cultured in Gibco Essential 8 medium (Thermo Scientific Fisher) on vitronectin coated cell culture plates. Differentiation into iPSC-derived cardiomyocytes was induced by the Wnt-signaling agonist CHIR99021 in combination with the GSK3 inhibitor IWP2 (both Sigma-Aldrich, St. Louis, MO, USA) as previously described by [26]. Electroporation of iPSC-derived cardiomyocytes (14 days after differentiation) was performed using the 4D Nucleofector system (program CA 137, Lonza, Cologne, Germany). 

### 2.6. Immunocytochemistry and Confocal Microscopy 

Cells were washed twice with phosphate buffered saline (PBS). Afterwards, they were fixed using 4% paraformaldehyde (Carl Roth, Karlsruhe, Germany) for 10 min at room temperature (RT). After washing with PBS, the cells were permeabilized using 0.1% Triton X-100 for 10 to 15 min at RT. 5% bovine serum albumin in PBS was used for unspecific blocking (30 min, RT). Primary anti-sarcomeric α-actinin antibodies (10 µg/mL, Sigma-Aldrich, St. Louis, USA) were used over night at 4 °C in combination with anti-mouse-IgG antibodies conjugated to Alexa488 (1:100, 1 h, RT, Thermo Fisher Scientific). The filamentous actin was stained with Alexa-Flour-488-phalloidin (Thermo Fisher Scientific) according to the manufacturer’s instructions. 4′,6-diamidin-2-phenylindole (DAPI, 1 µg/mL) was used for staining of the nuclei. After several washing steps with PBS, the cells were embedded in Vectashield Antifade Mounting Medium (Vector Laboratories, Burlingame, CA, USA) according to the manufacturer’s instructions. Confocal microscopy was done as previously described by [14]. 

### 2.7. Statistics

The statistical analysis was performed using the nonparametric Mann–Whitney test using GraphPad Prism V5.04 (GraphPad Software, San Diego, CA, USA). The indicated values are presented as mean ± standard derivation (SD). *P*-values < 0.05 were considered as significant. 

## 3. Results

At the age of 19 years, the male index patient (III-1, Figure 1A) received a routine cardiac check-up revealing biatrial enlargement and a right bundle branch block. He had no significant family history of cardiac disease, however, it was known that his parents were second cousins. The patient reported an active lifestyle, both during childhood and at the time of initial contact, participating in various forms of sport activities including weight lifting without cardiac symptoms. Between the ages of 21 and 22 years, he developed recurrent episodes of severe bradycardia due to a third degree atrioventricular (AV) block. In consequence, a permanent pacemaker was implanted. Subsequently, he was referred for an echocardiography to evaluate cardiac function as an underlying cardiomyopathy was suspected. The echocardiography revealed a normal sized left ventricle (LV) with mildly reduced systolic function (LV ejection fraction 45% ± 5%). Severe biatrial enlargement was present, with the left atrium (LA) volume at 166 mL/m^2^. The LV diastolic filling parameters showed the following: mitral valve (MV) E/A ratio 2.6, MV deceleration time 124 ms, LV E’ septal velocity 0.08 m/s, and LV E’ lateral velocity 0.25 m/s. The right atrial pressure (RAP) was measured at 20.00 mmHg. The LV diastolic filling parameters indicated severely abnormal diastolic filling and relaxation (E/A 2.6). Moreover, an elevated pulmonary artery systolic pressure and a markedly dilated inferior vena cava (IVC) with abnormal flow dynamics suggested right-sided pressure overload. In total, the clinical findings are in good agreement with the diagnosis of RCM in combination with AV conduction disease. A clinical follow-up at the age of 27 years, using echocardiography, confirmed the diagnosis of RCM (Figure 1B–D). The creatine kinase (CK) and B-type natriuretic peptide (NT-proBNP) values were highly increased (CK = 588.3 U/L and NT-proBN = 1240.5 pg/mL) supporting the diagnosis of heart insufficiency. Therefore, we screened III-1 for cardiomyopathy-associated genetic variants using a NGS panel of 174 genes associated with cardiomyopathy, which revealed a homozygous *DES* mutation, p.Y122H (Figure 1E). Several further nonsynonymous genetic variants were detected (Appendix A), which were mainly excluded as pathogenic mutations based on their minor allele frequency (MAF) above 0.01. Sanger sequencing was used for verification of the *DES* mutation, p.Y122H (Figure 1F). Although no detailed data from the parents are available, further family assessment indicated that only the paternal aunt (II-1, Figure 1A) developed a myopathy without cardiac involvement. Both grandmothers (I-2, 69 years and I-3, 64 years, Figure 1A) are reportedly sisters. Therefore, consanguinity within this family might be a likely explanation for homozygosity of the *DES* mutation, p.Y122H, found in the genome of III-1. Unfortunately, no DNA samples of other family members were available for genetic analyses. The *DES* mutation, p.Y122H, affects an amino acid at the beginning of coil 1, which is a hotspot for cardiomyopathy-associated mutations (Figure 1G). This tyrosine is highly conserved between several species (Figure 1H and Appendix A) and is part of the heptade sequence (d-position), which is highly important for the coiled-coil formation of desmin dimers (Figure 1I). In addition, several different bioinformatics prediction tools [27,28] indicate the deleterious impact of the *DES* mutation, p.Y122H (Table 1).

For the majority of pathogenic *DES* missense or small in-frame deletion mutations, an abnormal cytoplasmic aggregation persisting in the cardiomyocytes has been described [1,29,30]. Therefore, we constructed an expression plasmid for the *DES* mutation, p.Y122H, and for the previously reported *DES* mutation, p.Y122C, of an ACM-patient from the literature [22]. Cell transfection experiments, in combination with confocal microscopy, revealed, in two cell lines (HT-1080 and SW13) without any endogenous desmin expression for both mutations, severe aggregate formation, supporting their pathogenicity (Figure 2A,B). By contrast, the wildtype desmin formed regular intermediate filaments in the majority of the transfected cells (Figure 2A,B). For verification, we differentiated iPSCs from a healthy donor into cardiomyocytes and used these human iPSC-derived cardiomyocytes for cell transfection experiments using electroporation. Indeed, the *DES* mutation, p.Y122H, formed in the majority of transfected iPSC-derived cardiomyocytes, as well as abnormal cytoplasmic aggregates, whereas the wildtype desmin formed normal regular intermediate filaments (Figure 2C).

## 4. Discussion

Although the homozygous *DES* mutation, p.Y122H, was identified in an isolated index patient from a family without any obvious cosegregation, this mutation fulfills several criteria for pathogenicity according to the guidelines of the American College of Medical Genetics and Genomics (ACMG) [31]. First, the *DES* mutation, p.Y122H, is completely absent in the public Genome Aggregation Database (gnomAD, https://gnomad.broadinstitute.org, September 2019) [32], which is a moderate criterion for pathogenicity (PM2). In addition, several computational prediction tools indicate a deleterious effect of p.Y122H (supportive criterion PP3), as well, a different mutation, at the same tyrosine 122, has been recently described [22] (moderate criterion, PM5). This amino acid is localized at the highly conserved beginning of coil 1, which is a hotspot for cardiomyopathy-associated mutations (moderate criterion, PM1). Finally, our transfection experiments reveal functional evidence for its deleterious effect (strong criterion, PS3). In summary, the *DES* mutation, p.Y122H, fulfills one supportive, three moderate, and one strong criteria for pathogenicity. In consequence, the *DES* mutation, p.Y122H, has to be classified as a “pathogenic mutation” according to the ACMG guidelines [31].

In the 1990s, an abnormal desmin accumulation was described in the myocardial tissue from patients with RCM [33,34,35], however, it was not until 1998 that the first pathogenic *DES* mutations associated with RCM were described [36]. Since the original discovery by Goldfarb et al., some further RCM associated *DES* mutations have been described [8,37,38]. Of note, the clinical phenotypes of *DES* mutation carriers can change over time into other cardiomyopathy phenotypes making a correlation between genotype and clinical phenotype challenging [39]. In general, the clinical phenotypes associated with *DES* mutations are broad and include isolated skeletal myopathies, cardiomyopathies such as DCM, HCM, ACM, and NCCM, as well as combinations of both [1]. Interestingly, a second likely pathogenic *DES* mutation (p.Y122C), at the same amino acid position, has been previously described in a patient with ACM without a detailed further clinical description [22]. In our functional analyses, both variants at position 122 cause a severe filament assembly defect supporting their pathogenicity, however, these experiments do not explain why the associated clinical phenotypes are diverse and even appear at the heterozygous state. Nevertheless, it can be suggested that different genetic backgrounds, epigenetic differences, and presumably distinct environmental factors contribute to this phenomenon.

Desmin is a major muscle-specific intermediate filament protein. In addition to the conservation among different species, the beginning of the coil-1 domain is likewise highly homologous between desmin and further intermediate filament proteins, underlining the general functional relevance of this subdomain. Mutations in *GFAP*, encoding the glial fibrillary acidic protein, cause Alexander disease (MIM #203450), which is a disease of the central nervous system. Interestingly, Ye et al. described at the homologous *GFAP* position a comparable mutation (p.Y83H) in juvenile patients with Alexander disease [40]. Mutations in *LMNA*, encoding lamin A/C, cause cardiomyopathies, as well as Emery–Dreifuss muscular dystrophy (EDMD, MIM #181350). Of note, at the homologous tyrosine position 45 of lamin A/C, a pathogenic mutation (p.Y45C) has also been described in a sporadic case with EDMD [41]. These findings, in combination with our report on the *DES* mutation, p.Y122H, support the general functional relevance of this tyrosine for the intermediate filament family.

## 5. Conclusions

In summary, we describe and characterize a novel homozygous pathogenic missense mutation in *DES*, which is associated with RCM. This is in good agreement with other reports describing homozygous *DES* variants associated with RCM [37,42]. This is the first RCM-associated *DES* mutation within the highly conserved coil-1 region, leading to a severe filament assembly defect. Our report could be helpful for genetic counseling and it demonstrates that, in addition to autosomal dominant inheritance which is often combined with incomplete penetrance, sometimes recessive inheritance of *DES* mutations should also be considered, in particular in families where family history is difficult to assess or unclear.

## Figures and Tables

**Figure 1 genes-10-00918-f001:**
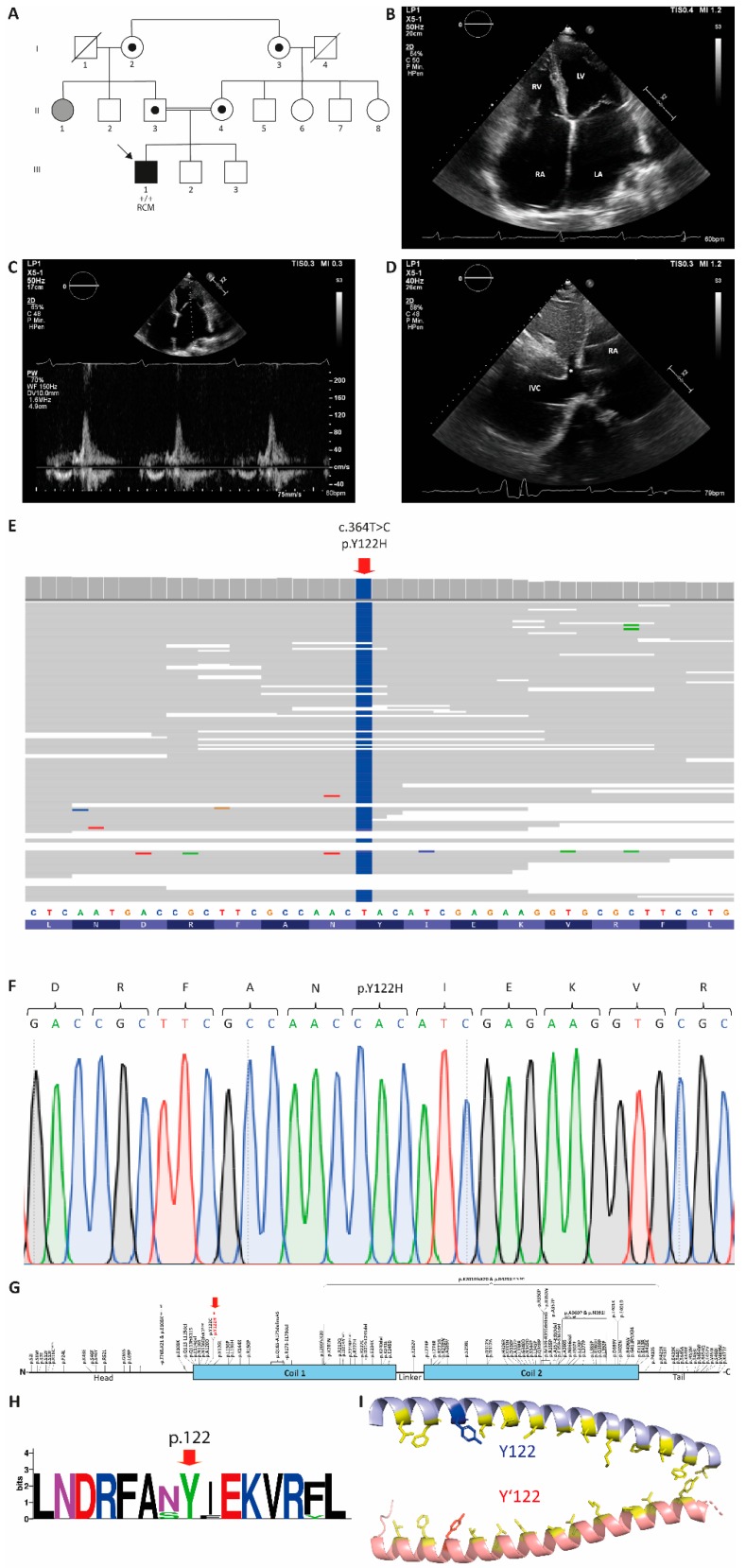
(**A**) Pedigree of the family. Circles represent females, squares represent males, and a slash denotes deceased. Black-filled symbols indicate a cardiac phenotype. Grey-filled symbols indicate individuals with skeletal myopathy. RCM, restrictive cardiomyopathy and +/+ indicates homozygous alleles. The index patient is marked with an arrow. Obligate carriers are indicated by small black circles. (**B**) Transthoracic echocardiography, four-chamber view. Note the enlarged size of both atria and the normal size of the ventricles. RA, right atrium; RV, right ventricle; LA, left atrium; and LV, left ventricle. (**C**) Transthoracic echocardiography, four-chamber view, pulsed wave Doppler of the mitral valve inflow. Note the restrictive flow pattern. (**D**) Transthoracic echocardiography, subcostal view. Note the enlarged inferior vena cava and hepatic vein (asterix). RA, right atrium and IVC, inferior vena cava. (**E**) Schematic overview of the next-generation sequencing analysis revealing p.Y122H. (**F**) Electropherogram of p.Y122H (III-1). (**G**) Schematic overview of the localization of known *DES* mutations within the desmin domain structure. Of note, p.Y122H (red arrow) is localized in a genetic hotspot of cardiomyopathy-associated *DES* mutations at the beginning of the coil-1 domain. (**H**) Partial sequence logo of desmin. Of note, p.Y122 is highly conserved (red arrow). For the complete alignment, see Appendix A. (**I**) Desmin dimer, modeled using Swiss-Model (PDB ID: 3S4R; [24]). Mutant tyrosine residues (p.Y122 and p.Y122’) are highlighted in blue and red. Residues of the heptad sequence are indicated in yellow.

**Figure 2 genes-10-00918-f002:**
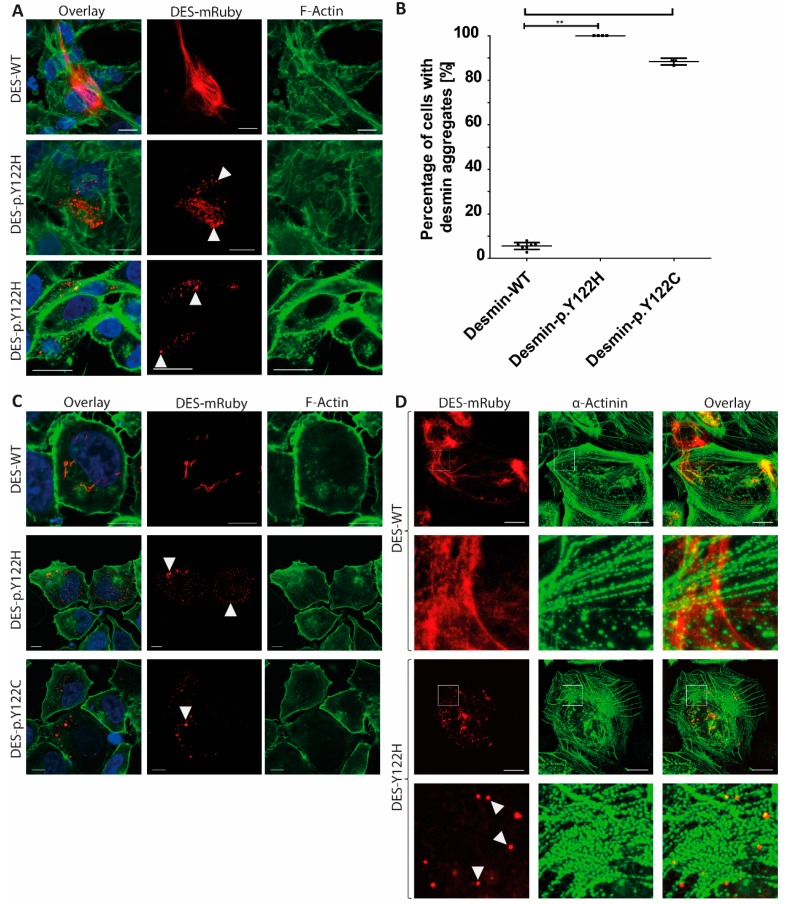
(**A**) Representative confocal microscopy images of transfected HT-1080 cells. The fluorescence intensity of wildtype and mutant desmin mRuby is shown in red. The fluorescence intensity of Alexa-488 conjugated to phalloidin (for F-actin staining) is shown in green and the nuclei were stained using DAPI (blue). (**B**) The statistical analysis of desmin aggregate formation revealed for both mutations (p.Y122H and p.Y122C) a severe aggregate formation in the majority of transfected HT-1080 cells. ** *p*-value < 0.01. (**C**) Representative confocal microscopy images of transfected SW13 cells. (**D**) Confocal microscopy of transfected iPSC-derived cardiomyocytes. The fluorescence intensity of wildtype and mutant desmin mRuby is shown in red. Alpha-actinin, as a cardiomyocytes marker, was stained using primary and secondary antibodies conjugated to Alexa488 (green). Scale bars represent 25 µm.

**Table 1 genes-10-00918-t001:** Overview about different prediction tools analyzing the desmin (*DES)* mutation p.Y122H.

PolyPhen-2 ^1^	Mutation Taster ^2^	Provean ^3^	Panther ^4^	SNPs & GO ^5^	SIFT
Probably damaging 0.999	Disease causing 83	Deleterious −4.228	Probably damaging 750	Disease-associated variant 0.613	Deleterious

^1^http://genetics.bwh.harvard.edu/pph2/, ^2^http://www.mutationtaster.org, and ^3^http://provean.jcvi.org; ^4^http://www.pantherdb.org; ^5^http://snps.biofold.org.

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
