# Peer review of "Restrictive Cardiomyopathy is Caused by a Novel Homozygous Desmin (DES) Mutation p.Y122H Leading to a Severe Filament Assembly Defect"

_genes, 2019, doi:10.3390/genes10110918_

Round 1

Reviewer 1 Report

Comments to the authors:

In this paper Brodehlet and colleagues present a patient with restrictive cardiomyopathy carrying a homozygous missense mutation in DES gene (c.364T>C; p.Y122H). Using functional assays (plasmid expression in HT-1080 cells and in iPSCs) they gave evidence of the deleterious effect of this variant.

There are only minor comments:

Results

page 4, line 143. There is a discrepancy between the text and the figure 1A. In the text the aunt with the myopathy is maternal while in the figure is paternal. Please correct.

Discussion.

Page 7, line 200. According to the ACMG guidelines PM4 is “Protein length changes as a result of in-frame deletions/insertions in a non-repeat region or stop-loss variants” while the authors refer to PM5 (“Novel missense change at an amino acid residue where a different missense change determined to be pathogenic has been seen before”). Please correct. The authors do not consider other cases with RCM and homozygous variants in DES (e.g. PMID 16376610; PMID 19433360) previously described in literature. Why? They should consider the possibility to cite them in the discussion/ conclusion.

Author Response

Reviewer #1-1

In this paper Brodehl and colleagues present a patient with restrictive cardiomyopathy carrying a homozygous missense mutation in DES gene (c.364T>C; p.Y122H). Using functional assays (plasmid expression in HT-1080 cells and in iPSCs) they gave evidence of the deleterious effect of this variant. There are only minor comments: Results page 4, line 143. There is a discrepancy between the text and the figure 1A. In the text the aunt with the myopathy is maternal while in the figure is paternal. Please correct.

Answer of the reviewers

We apologize for this mistake and change the text accordingly.

Reviewer #1-2

Discussion. Page 7, line 200. According to the ACMG guidelines PM4 is “Protein length changes as a result of in-frame deletions/insertions in a non-repeat region or stop-loss variants” while the authors refer to PM5 (“Novel missense change at an amino acid residue where a different missense change determined to be pathogenic has been seen before”). Please correct.

Answer of the reviewers

We thank reviewer for mentioning this point and changed the revised manuscript according to this suggestions of reviewer #1.

Reviewer #1-3

The authors do not consider other cases with RCM and homozygous variants in DES (e.g. PMID 16376610; PMID 19433360) previously described in literature. Why? They should consider the possibility to cite them in the discussion/ conclusion.

Answer of the reviewers

We agree to reviewer #1 and cite the relevant manuscripts in the revised version of the Manuscript.

Reviewer 2 Report

This study reports a novel mutation in the desmin gene underlying a recessive form of restrictive cardiomyopathy. Although the mutation is novel, the study could benefit from alterations and further functional studies. I have the following comments:

Major

Why did the authors choose to use HT-1080 cells (a fibrosarcoma line) as opposed to AT-1 or HL-1 cells, which are the most commonly used cell lines as they are derived from heart samples?

The authors used wildtype iPSC -CMs, which they subsequently artificially transfected with a plasmid to express a mutant form of the protein. Yet, these cells express wildtype desmin too. This model, accordingly, does not recapitulate a recessive disease. The authors should use iPSC- CMs derived from the homozygous patient himself.

The fact that the patient carries several other mutations in several heart genes, some of which appear to be pathogenic too, adds further support to the necessity to perform experiments on iPSC- CMs derived from him.

Since the authors constructed a plasmid for the ACM-causing Y122C mutation, they should perform experiments that show the phenotypic difference between the two variants and hence explain why one causes RCM while the other causes ACM. For instance, is plakoglobin misplaced from the junctions? Is GSK3β translocating to the intercalated disks? Is there gap junction remodelling?

Minor

Introduction lines 48-49: please spell out the gene names in addition to their abbreviations

Introduction: The authors refer our inability to explain why certain mutations cause different cardiomyopathies. It would be useful to comment on why certain mutations cause skeletal myopathy as opposed to heart disease too. Also – not all desmin mutations cause aggregates. There are several cases of cardiomyopathies underlined by DES mutations where no aggregates of the intermediate filaments are found in the heart.

Author Response

Reviewer #2-1

Why did the authors choose to use HT-1080 cells (a fibrosarcoma line) as opposed to AT-1 or HL-1 cells, which are the most commonly used cell lines as they are derived from heart samples? The authors used wildtype iPSC -CMs, which they subsequently artificially transfected with a plasmid to express a mutant form of the protein. Yet, these cells express wildtype desmin too. This model, accordingly, does not recapitulate a recessive disease. The authors should use iPSC-CMs derived from the homozygous patient himself.

Answer of the reviewers

We thank reviewer #2 for this question. To address the questions of reviewer #2, we repeated the experiments in SW13 cells. This cell line does not express any cytoplasmic intermediate filaments [1] and is in consequence frequently used to investigate DES mutations to prevent interference with any endogenously expressed cytoplasmic intermediate filament proteins [2, 3]. We incorporated these data into the revised version of the manuscript (Figure 2C) and updated the Material & Methods paragraph according to the additional experiments. We agree to reviewer #2 that HL-1 cells are also frequently used in cardiac research. However, we want to underline that HL-1 express also endogenous desmin [2]. However, HL-1 cells are murine cells [4], which mimic only partially cardiomyocytes [5]. The advantages of HL-1 cells as an alternative to human cardiomyocytes derived from induced pluripotent stem cells remain still unclear to us. Unfortunately, we do not have primary cells of the index patient carrying DES-p.Y122H to generate iPSCs. Therefore, we have used iPSC-derived cardiomyocytes from a healthy donor and transfected these cardiomyocytes with plasmids encoding wild-type and mutant desmin.

 Reviewer #2-3

The fact that the patient carries several other mutations in several heart genes, some of which appear to be pathogenic too, adds further support to the necessity to perform experiments on iPSC- CMs derived from him.

Answer of the authors

We think that this is a misunderstanding of reviewer #2. All other variants were classified according to the ACMG guidelines [6] as variants of unknown significance (VUS). To be more precise and prevent any misunderstanding of the readers, we updated Table S1 by inserting the ACMG classification for each variant.

Reviewer #2-4

Since the authors constructed a plasmid for the ACM-causing Y122C mutation, they should perform experiments that show the phenotypic difference between the two variants and hence explain why one causes RCM while the other causes ACM. For instance, is plakoglobin misplaced from the junctions? Is GSK3β translocating to the intercalated disks? Is there gap junction remodelling?

Answer of the authors

It is well accepted that the associated genotypes, including the DES gene, overlap between the different clinical cardiomyopathies [7]. As we have mentioned in our manuscript, this phenomenon can even occur sometimes within the same family [8]. As we have mentioned in our manuscript the mutation DES-p.Y122C has been reported in an ACM patient by Walsh et al. [9]. However, because this is not our patient we do not have cardiac tissue to perform the suggested experiments of reviewer #2. In addition, the plakoglobin and GSK3β negative staining experiments have been reported as specific findings in ACM patients and the relevance for restrictive cardiomyopathy (RCM) is currently completely unclear [10, 11]. In contrast to the report of Walsh et al. our described index patient (III-1) presented RCM and currently he is not heart transplanted. In this context, the relevance and impact of the suggested staining experiments are unclear to us. Therefore, we have not changed the revised version of our manuscript according to this point.

Reviewer #2-5

Introduction lines 48-49: please spell out the gene names in addition to their abbreviations.

Answer of the authors

We thank reviewer #2 for this point and changed the manuscript accordingly.

Reviewer #2-6

Introduction: The authors refer our inability to explain why certain mutations cause different cardiomyopathies. It would be useful to comment on why certain mutations cause skeletal myopathy as opposed to heart disease too.

Answer of the authors

We inserted this point into the introduction of the revised manuscript.

Reviewer #2-7

Also – not all desmin mutations cause aggregates. There are several cases of cardiomyopathies underlined by DES mutations where no aggregates of the intermediate filaments are found in the heart.

Answer of the authors

We agree with reviewer #2 to this point. Different other pathomechanisms like altered nano-mechanical properties [12] or altered protein-protein interactions might be involved. Therefore, we updated the introduction to this point.

Additional References

Sarria, A. J.; Lieber, J. G.; Nordeen, S. K.; Evans, R. M., The presence or absence of a vimentin-type intermediate filament network affects the shape of the nucleus in human SW-13 cells. Journal of cell science 1994, 107 ( Pt 6), 1593-607.

Brodehl, A.; Hedde, P. N.; Dieding, M.; Fatima, A.; Walhorn, V.; Gayda, S.; Saric, T.; Klauke, B.; Gummert, J.; Anselmetti, D.; Heilemann, M.; Nienhaus, G. U.; Milting, H., Dual color photoactivation localization microscopy of cardiomyopathy-associated desmin mutants. The Journal of biological chemistry 2012, 287, (19), 16047-57.

Bar, H.; Fischer, D.; Goudeau, B.; Kley, R. A.; Clemen, C. S.; Vicart, P.; Herrmann, H.; Vorgerd, M.; Schroder, R., Pathogenic effects of a novel heterozygous R350P desmin mutation on the assembly of desmin intermediate filaments in vivo and in vitro. Human molecular genetics 2005, 14, (10), 1251-60.

Claycomb, W. C.; Lanson, N. A., Jr.; Stallworth, B. S.; Egeland, D. B.; Delcarpio, J. B.; Bahinski, A.; Izzo, N. J., Jr., HL-1 cells: a cardiac muscle cell line that contracts and retains phenotypic characteristics of the adult cardiomyocyte. Proceedings of the National Academy of Sciences of the United States of America 1998, 95, (6), 2979-84. White, S. M.; Constantin, P. E.;

Claycomb, W. C., Cardiac physiology at the cellular level: use of cultured HL-1 cardiomyocytes for studies of cardiac muscle cell structure and function. American journal of physiology. Heart and circulatory physiology 2004, 286, (3), H823-9.

Richards, S.; Aziz, N.; Bale, S.; Bick, D.; Das, S.; Gastier-Foster, J.; Grody, W. W.; Hegde, M.; Lyon, E.; Spector, E.; Voelkerding, K.; Rehm, H. L.; Committee, A. L. Q. A., Standards and guidelines for the interpretation of sequence variants: a joint consensus recommendation of the American College of Medical Genetics and Genomics and the Association for Molecular Pathology. Genetics in medicine : official journal of the American College of Medical Genetics 2015, 17, (5), 405-24.

Brodehl, A.; Ebbinghaus, H.; Deutsch, M. A.; Gummert, J.; Gartner, A.; Ratnavadivel, S.; Milting, H., Human Induced Pluripotent Stem-Cell-Derived Cardiomyocytes as Models for Genetic Cardiomyopathies. International journal of molecular sciences 2019, 20, (18).

Bergman, J. E.; Veenstra-Knol, H. E.; van Essen, A. J.; van Ravenswaaij, C. M.; den Dunnen, W. F.; van den Wijngaard, A.; van Tintelen, J. P., Two related Dutch families with a clinically variable presentation of cardioskeletal myopathy caused by a novel S13F mutation in the desmin gene. European journal of medical genetics 2007, 50, (5), 355-66.

Walsh, R.; Thomson, K. L.; Ware, J. S.; Funke, B. H.; Woodley, J.; McGuire, K. J.; Mazzarotto, F.; Blair, E.; Seller, A.; Taylor, J. C.; Minikel, E. V.; Exome Aggregation, C.; MacArthur, D. G.; Farrall, M.; Cook, S. A.; Watkins, H., Reassessment of Mendelian gene pathogenicity using 7,855 cardiomyopathy cases and 60,706 reference samples. Genetics in medicine : official journal of the American College of Medical Genetics 2017, 19, (2), 192-203.

Asimaki, A.; Tandri, H.; Huang, H.; Halushka, M. K.; Gautam, S.; Basso, C.; Thiene, G.; Tsatsopoulou, A.; Protonotarios, N.; McKenna, W. J.; Calkins, H.; Saffitz, J. E., A new diagnostic test for arrhythmogenic right ventricular cardiomyopathy. The New England journal of medicine 2009, 360, (11), 1075-84.

Chelko, S. P.; Asimaki, A.; Andersen, P.; Bedja, D.; Amat-Alarcon, N.; DeMazumder, D.; Jasti, R.; MacRae, C. A.; Leber, R.; Kleber, A. G.; Saffitz, J. E.; Judge, D. P., Central role for GSK3beta in the pathogenesis of arrhythmogenic cardiomyopathy. JCI insight 2016, 1, (5).

Kreplak, L.; Bar, H., Severe myopathy mutations modify the nanomechanics of desmin intermediate filaments. Journal of molecular biology 2009, 385, (4), 1043-51.

Round 2

Reviewer 2 Report

I thank the authors for responding to each one of the points raised and for altering the manuscript accordingly. Also for performing additional experiments on a different cell line. I do not have any further comments. I believe that in its current format the manuscript merits publication.